# Registered Data-Centered Lab Management System Based on Data Ownership Safety Architecture

Xuying Zheng [1,2], Fang Miao [3], Piyachat Udomwong [1,*] and Nopasit Chakpitak [1,*]

1   International College of Digital Innovation, Chiangmai University, Chiangmai 50200, Thailand;
    xuying_zheng@cmu.ac.th
2   School of Architecture and Civil Engineering, Chengdu University, Chengdu 610106, China
3   Big Data Research Institute, Chengdu University, Chengdu 610106, China
*   Correspondence: piyachat.u@cmu.ac.th (P.U.); nopasit@cmuic.net (N.C.)

**Abstract:** University and college laboratories are important places to train professional and technical personnel. Various regulatory departments in colleges and universities still rely on traditional laboratory management in research projects, which are prone to problems such as untimely information and data transmission. The present study aimed to propose a new method to solve the problem of data islands, explicit ownership, conditional sharing, data safety, and efficiency during laboratory data management. Hence, this study aimed to develop a data-centered lab management system that enhances the safety of lab data management and allows the data owners of the labs to control data sharing with other users. The architecture ensures data privacy by binding data ownership with a person using a key management method. To achieve data flow safely, data ownership conversion through the process of authorization and confirmation was introduced. The designed lab management system enables laboratory regulatory departments to receive data in a secure form by using this platform, which could solve data sharing barriers. Finally, the proposed system was applied and run in different server environments by implementing data security registration, authorization, confirmation, and conditional sharing using SM2, SM4, RSA, and AES algorithms. The system was evaluated in terms of the execution time for several lab data with different sizes. The findings of this study indicate that the proposed strategy is safe and efficient for lab data sharing across domains.

**Keywords:** university laboratory management; data sharing; data ownership safety architecture; conditional sharing; safety and efficiency





## 1. Introduction

The importance of knowledge for organizations is now widely recognized, being one of the resources whose management influences the success of organizations through the exchange and sharing of information, knowledge, and experience among its members. Numerous challenges are associated with the management of laboratory data as labs generate a lot of experimental and management data on daily basis. Frequent experimental accidents and the leakage of hazardous chemicals is worthy of attention. Supervision departments, such as the Education Bureau, Emergency Bureau, and Public Security Department, need various types of laboratory management data to report while supervising laboratory safety.

At present, existing laboratory management systems used by each university, regulatory department, and even various laboratories are different, which leads to obvious problems when lab data collection or emergency response is needed. For example, which laboratory management data are classified confidential, and which data can be handed over to relevant departments? Who must claim the ownership of laboratory-related data? Which departments can view or use the data? Could lab data be transferred to other departments? These questions raised laboratory data ownership issues. Secondly, some data that relate to laboratory management need to be kept confidential. Where and how can we secure this

data safely? Is it safe for different departments to transmit data? How can we guarantee that there will be no data transfer to others? These can be summarized as data-safe storage and transmission questions. In recent years, frequent university laboratory accidents have occurred; in such cases, emergency responding units and other regulatory departments need to collect real-time lab-related data. If it is still a regular report retrieval process, it cannot meet the requirements of time efficiency. So, we need to find efficient and trustworthy method to solve this problem of obtaining data in a timely manner.

The main reason for the above problems is that the laboratory management data ownership is unclear and the existing lab management information systems (LMIS) of each branch are independent. It is impossible to collect a real-time, tamper-free, statistically accurate data when it is needed. To deal with these problems, we intend to adopt data ownership safety architecture (DOSA) to safely circulate laboratory management data and cultivate an ecological and sustainable forest from the "flowerpot" of each independent system [1].

Laboratories are currently using lab management information systems frequently (LIMS) [2]. A university may have multiple management systems, including an equipment management system, a chemical management system, a safety examination system, etc. Related systems comprise relevant departments of the laboratory, such as the experiment management department of the Education Bureau, the accident handling department of the Emergency Bureau, and the hazardous chemicals supervision department of public security. Such complicated systems make it difficult to obtain information accurately and quickly when the same report requires the cooperation of different departments or laboratories. Compared to traditional manual records, the lab management information system generated by a specific business unit or company can be easily outdated and has high maintenance costs. Once the database needs huge updates related to its data source, the existing system cannot meet the demand, and a new system must be designed so that all data can be transferred, resulting in increased costs. The linkage between different systems is weak; the efficiency is relatively low; and the security is not uniform and difficult to guarantee when it is necessary to submit data.

The medical industry uses the blockchain system to manage medical imaging data, and the banking system uses blockchain as a new type of financial technology [3]. Blockchain has high energy consumption, expensive development costs, but relatively high security. At present, some scholars have produced a framework diagram of the blockchain management in university education, but there is no corresponding technical means [4].

Considering the problems mentioned above, the lab industry would require a technology that allows data to flow securely and efficiently. This article's contributions are as follows: Experiment-related data management is based on the data ownership structure. It designs a security technology so that data ownership binds with a person and it can be safely registered on this platform; meanwhile, data can determine ownership and data users and owners can conditionally share or trade using ownership conversion across domains and across borders efficiently and safely. Our proposed laboratory data management architecture is shown in Figure 1.

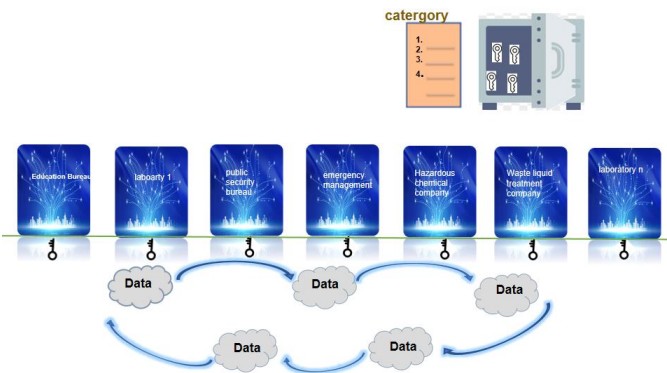

**Figure 1.** Proposed data ownership safety architecture to manage the laboratory.

The current study has been organized as follows: Section 2 explains related theories and key management. Section 3 presents the core strategies including laboratory data encrypted registration and an authorized transfer method for conditional sharing. Section 4 details the whole lab management framework and the incident searching case study based on DOSA using the above ownership conversion method. Section 5 verifies the effectiveness of the data flow process using key management based on proposed core strategies. Finally, Section 6 summarizes the paper.

## 2. Related Work

### 2.1. Data Ownership Safety Architecture

As data cannot be safely circulated and shared, Miao et al. proposed a data ownership security architecture that contains one body with two wings [5]. The body represents data binding ownership: one wing is the key; the other wing means value.

To solve the problem of data transfer barriers, DOSA proposes a global, cross-border, comprehensive, collaborative, safe, and efficient data architecture system. The whole field is no longer the cross-domain data of the head industry or a single system as previous, but all industry-related data are registered. Integration and collaboration mean that data are retrieved and used by different industry sources. This architecture has efficient security technologies for data security from source, registration, circulation, and use to supervision.

DOSA is composed of a data register center (DRC), a data authorization center (DAC), a data exception center (DEC), a data application units (DAUs), a key system, and other components. Safety and efficiency are the core of DOSA.

#### 2.1.1. DRC

The data register center stores the data in an encrypted form, which is invisible even to the administrator. It not only confirms the ownership of the data but also keeps it safe and confidential. After the data are registered, the directory is automatically generated, which is convenient for data users to view and find relevant data sources. It can register the data of relevant units or laboratories to DRC or link to DRC through API [6].

#### 2.1.2. DAC

To ensure the relationship between people and data ownership, DAC is responsible for the process of data ownership confirmation and authorization [7]. By guaranteeing the interests of data owners, they are confident to put their data on the platform. The owner of the data divides data-related personnel into data owners, data producers, and data users. In the current study, a design was proposed to save all the related public keys to the DAC, and the private key is kept by their client, without placing it on any platform or cloud, to ensure security.

#### 2.1.3. DEC

To protect the interests of data owners, DEC needs to supervise during the data flow process. From the point of data register and data conditional sharing, even after the trading process, we obtained blockchain technical means, such as watermarks and timestamps, and added them to the data authorization process to ensure data users cannot transmit again without the owner's permission [8].

DOSA has designed some framework applications in tourism, lab education, and data product trading processes [9,10].

### 2.2. Laboratory Management Technology

Existing laboratory management usually uses a laboratory information management system (LIMS) that has collected related data through various laboratories. Some benefits of using LIMS systems are the ability to track individual data items or samples from reported results [11]. Some limitations of LIMSs for this specific application are limited, specialized for specific equipment, or applicable to chemistries [12]. Some LIMS chemistry systems

are only available to connect with the department of public safety. It is rare to find a LIMS system for chemistry widely used in related departments. There is also a high price for many commercial LIMS systems. Various systems' costs are expensive for labs, and customization is limited to what the system offers [13]. Additionally, experimental data are private to external parties, but it needs to be shared among the research team.

Recently, researchers have designed some LIMSs for education or other specific applications. Focusing on education, there is interactive laboratory information management system, which is designed for experimental teaching management, experimental equipment management, laboratory open management, communication, and interactive management modules [14]. For specific disciplines, there is an information management system specifically designed for high-level biosafety labs or biobanks [15,16]. However, these systems only flow data inside the lab, in which data cannot flow quickly to supervision departments or related companies. Some researchers realized that the LIMS needs both safety and efficiency in order to enable laboratory staff to conduct their research as freely as possible; therefore, the information on the research contents, laboratory personnel, experimental materials, and experimental equipment was collected and utilized by only one system [17]. However, the current research mainly focuses on the system design, and there is no specific experimental technical means to verify it. Thus, we employed DOSA to manage lab data to safely and quickly flow data inside or outside the laboratory. We have compared the proposed architecture to existing lab management system in Table 1.

**Table 1.** Comparison of the proposed architecture and existing lab management.

|  | **Proposed DOSA Lab Management** | **Existing Lab Management System** |
|---|---|---|
| Data Ownership | Clear | Undefined |
| Data security | The public key in DAC | Key pair in systems |
| Data source | Global | Specific service [18] |
| Data register | Encrypted, fully automatic | Admin visible, semi-automatic entry |
| Data Search | Authorization visible | Collect systems to summarize |
| Application | for a variety of businesses | For a single item [19] |

However, DOSA-related studies have designed models on some domains such as tourism, which have only frameworks without implementation [9]. We propose to use technical methods to solve data ownership by binding persons that used a key management system to protect data privacy.

### 2.3. Key System Method

To ensure data registration and sharing security, this paper adopted RSA and AES and Chinese domestic commercial keys SM2 and SM4 to compare the aspects of safe and efficiency. Table 2 summarizes the pros and cons of these key algorithms.

**Table 2.** Comparison of the four key algorithms.

| Algorithm | Description | Advantage | Disadvantage |
|---|---|---|---|
| **RSA** | Asymmetric encryption that encrypts with public key; decrypts with private key [20]. | In the process of encryption and decryption, there is no need to transmit confidential keys through the network. The key management is better than the AES algorithm. | The speed of encryption and decryption is relatively slow, usually it is not suitable for the encryption of a large amount of data files. |
| **AES** | Symmetric encryption algorithm. Encryption and decryption processes use the same set of keys. | The operation does not require a computer with very high processing power and large memory. The operation resists attacks easily. It always maintains good performance in different operating environments; the encryption speed is relatively fast. | It is required to secretly distribute the key before communication. The decrypted private key must be transmitted to the receiver of the encrypted data through the network. |

**Table 2.** *Cont.*

| Algorithm | Description | Advantage | Disadvantage |
|---|---|---|---|
| SM2 | Asymmetric encryption algorithm which is an elliptic curve public key cryptography algorithm based on ECC [21]. | Compared with RSA, the performance of SM2 is better and more secure. The password complexity is high, the processing speed is fast, and the computer performance consumption is comparatively small. | Encryption and decryption take a relatively long time and are suitable for encrypting small amounts of data. |
| SM4 | Symmetric encryption algorithm, the key length and block length are both 128 bits. | Safe and efficient, easy to implement software and hardware. It usually has a fast computing speed. | The management and distribution of keys are relatively difficult and not secure enough. |

SM2 and SM4 are cryptographic standards authorized to be used in China. Relevant studies have shown that the SM2 and SM4 algorithms are more secure than ECDSA and AES [22]. During a controlled experiment, AES outperforms SM4 by a significant margin [23]. Symmetric encryption is usually used when the message sender needs to encrypt a large amount of data. It has the characteristics of an open algorithm, a small amount of calculation, and fast encryption speed. The advantage of the symmetric encryption algorithm lies in the high speed of encryption and decryption, and the difficulty of cracking when using a long key. The disadvantages of symmetric encryption are that key management and distribution are difficult and insecure. Before the data are transmitted, the sender and the receiver must agree on a secret key, and both parties must keep the key. If the key of one party is leaked, the encrypted information is insecure, and the security cannot be guaranteed [24].

The advantage of asymmetric encryption is its higher security. The public key is made available, and the private key is kept by itself so there is no need to give the private key to others. The disadvantage of asymmetric encryption is that its speed is relatively slow, so it is only suitable for encrypting a small amount of data.

## 3. Methodology

### 3.1. Laboratory-Related Management Data and Persons

Laboratory management data are divided into unconditional and conditional sharing. For example, the opening hours of the laboratory are unconditional sharing data. The data transmission of the inventory and usage of hazardous chemicals needs conditional sharing by the administrative department (data owners). Safety learning data, accident statistics, and related teaching video data are transnational conditional sharing data. Lab-related data are classified into 0, 1, 2, 3, 4, and 5 according to the security level [25]. Class 0 means data can be public, such as laboratory introductions and research group members' profiles which do not need any security protection. Class 5 generally is a national key scientific research experiment whose data need to be kept strictly confidential. Class 1 to 4 refers to the data that the laboratory needs to share or that are trade-protected.

Lab data producers can be one or several persons, such as laboratory administrators, experimental instructors, or relevant teachers. A data owner is usually a person or a unit, such as a person in charge of the laboratory team or the emergency department. Data producers can input data, but they do not have the authority to share or trade data. After the data user has found targeted data by querying the DRC category, they need to contact the data owner. If the data owner considers that the data user can apply to view or use these data sources, the owner finds the user's public key from the DAC then encrypts these data with the user's public key to complete the authorization. The data user opens encrypted lab data with his private key to finish the conditional sharing of data (Figure 2a,b).

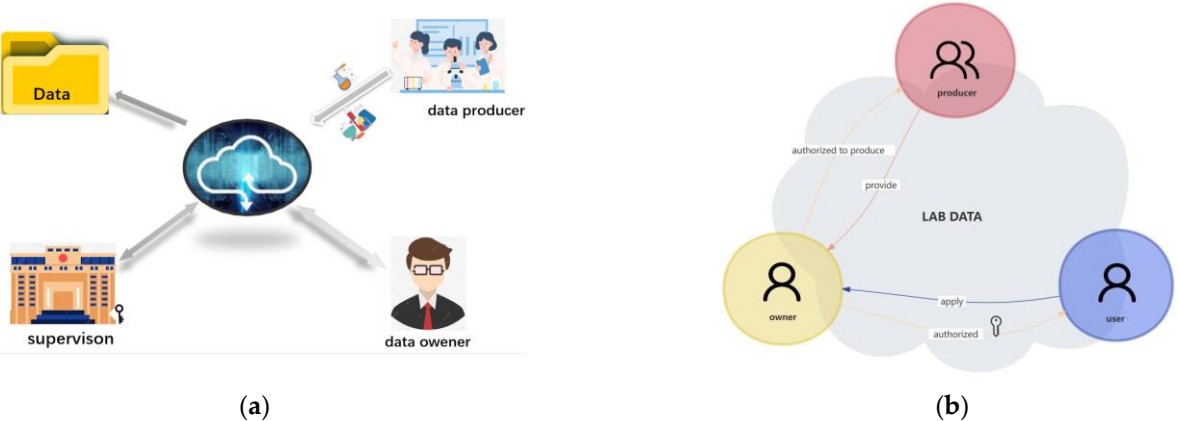

| (**a**) | (**b**) |

**Figure 2.** Illustrates the relationship of data-related persons or branches (**a**) and the relationship between data user, producer, and owner (**b**).

### 3.2. Data Encryption Registration and Application Process

The laboratory data are uploaded to the DRC by the owner, and the owner's public key is added to verify the ownership of the data. It can guarantee the privacy and security of the data at the source. The data are stored in the DRC, which the administrator cannot view without permission. The encrypted lab data are transmitted to the DRC, meanwhile the directory is automatically generated by keywords. The relevant users can apply for permission to obtain the data after searching in the data directory. Figure 3 shows the data owner encrypting the registration data, which automatically generates an index directory, then data users apply for the required data by viewing the catalog.

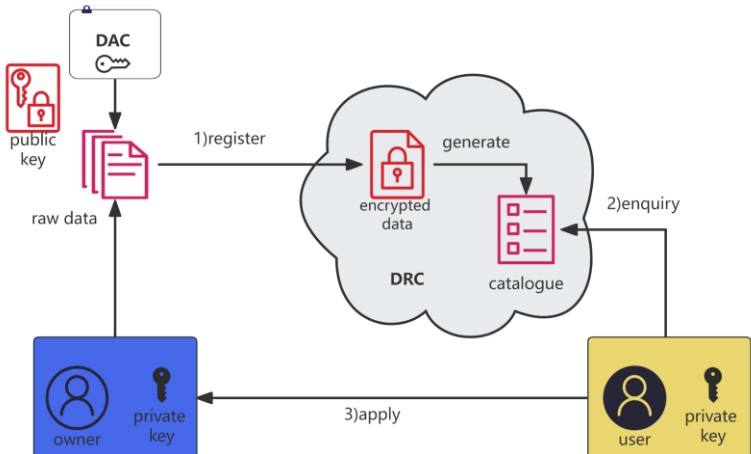

**Figure 3.** Data encryption registration process.

### 3.3. Confirmation and Authorization of the Data Transmission Process

3.3.1. Mutual Trust—Single-Layer Encryption

After the data user searches the DRC catalog and queries the data they need, the data user communicates with the data owner. If the user obtains permission, the data owner encrypts the data with the user's public key. Then, the data user can use his private key to decrypt data. Meanwhile, to ensure the interests of the data owner, the entire process has traceability technology in the DOSA lab management system. For example, time stamps and digital signatures are adopted, which can ensure that data users do not transmit data outside [26]. If the mutual parties trust each other, the process of conditional sharing occurs, as shown in Figure 4. Table 3 shows the definition of symbols used. Algorithm 1 shows how the lab data user retrieves data in the DRC during the process of data ownership conversion.

| Algorithm 1. Encryption_Once ( ) |
|---|

**Input:** Lab data owner $O_n$ with their their lab data record $R_n$. Lab data user with their private key $Uprk_n$.
**Output:** Boolean(True or False)

| | |
|---|---|
| 1. | #Function used to encrypt the lab data record. |
| 2. | **For** owner O query the user's public key $Upubk_n$ from the DAC $Dacv_n$ |
| 3. | #Check the DAC's data |
| 4. | **if**(role == "Owner") then |
| 5. | Encryption with the user's public key $Upubk_n$ |
| 6. | lab data record $R_n \rightarrow$ encrypted data |
| 7. | encrypted data $\rightarrow Drc_n$ |
| 8. | **return** True |
| 9. | **else** |
| 10. | **return** False |
| 11. | **end if** |
| 12. | **end for** |
| 13. | **For** user $U$ query the data register center catalogue view $Drcv_n$ |
| 14. | **if**(role == "User") then |
| 15. | Decryption using the user's private key $Uprk_n$ |
| 16. | $Drc_n \rightarrow$ encrypted data |
| 17. | encrypted data $\rightarrow$ lab data record $R_n$ |
| 18. | **return** True |
| 19. | **else** |
| 20. | **return** False |
| 21. | **end if** |
| 22. | **end for** |
| 23. | **end** function |

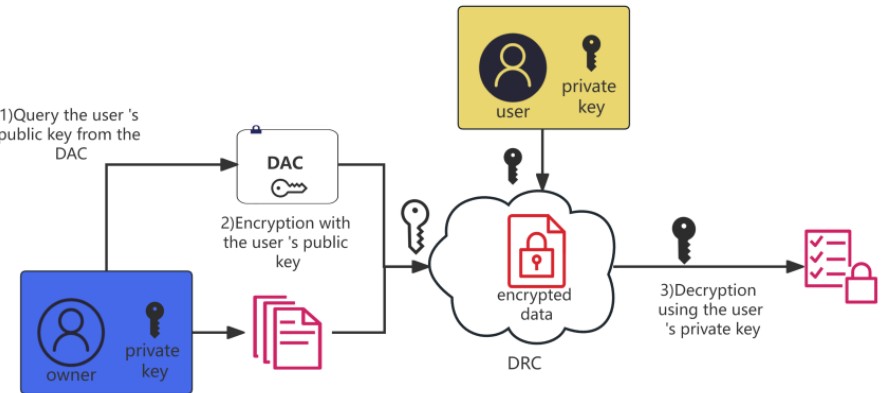

**Figure 4.** Single-layer encryption.

**Table 3.** The definition of symbols used.

| Symbols | Definition |
|---|---|
| $U_n$ | $n$th Lab Data User |
| $O_n$ | $n$th Lab Data Owner |
| $R_n$ | $n$th Lab Data Record |
| $Upubk_n$ | $n$th User Public Key |
| $Uprk_n$ | $n$th User Private Key |
| $Opub_n$ | $n$th Owner Public Key |
| $Oprk_n$ | $n$th Owner Private Key |
| $Drc_n$ | $n$th Data Register Center |
| $Drcv_n$ | $n$th Data Register Center Catalogue View |
| $Dacv_n$ | $n$th Data Authority Center View |

### 3.3.2. Data Users Do Not Trust the Data Source—Double Encryption

After checking the catalog in the DRC and communicating with the data owner about the data product's price, the data user may doubt the authenticity of the data source and ownership. In this situation, the data owner will add this data product to the owner's private key, then encrypt it by using the user's public key, which is found in the DAC. As the next step, data owners input this encrypted data to the DRC and then inform the users.

Firstly, the data user finds the data owner's public key from the DAC, and they use it to open the double-encrypted data product. As the next step, the data user utilizes his private key to decrypt the data. Figure 5 shows the transaction process of experimental data products when the data user questions the data authenticity or the data source's ownership.

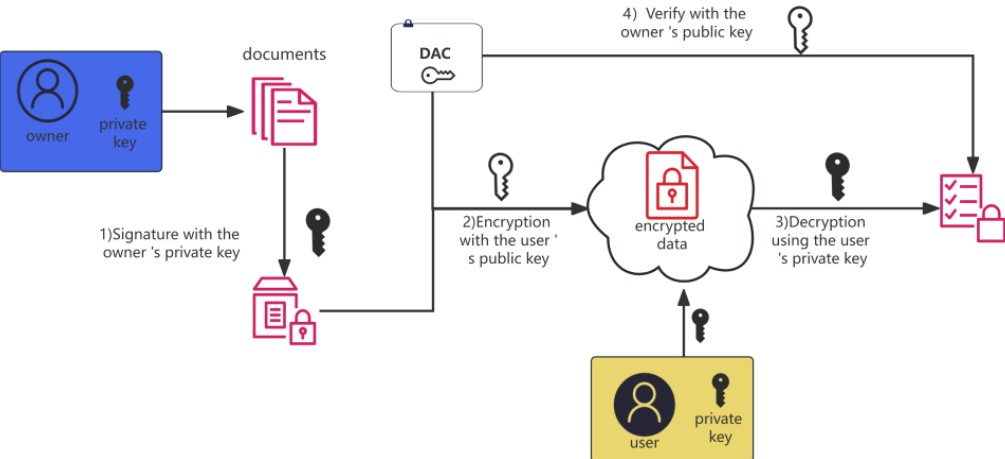

**Figure 5.** Double encryption.

Using a person-binding data owner method to finish data conditional sharing can break data barriers. We also resolve issues of the laboratory data class and ownership conversion process.

## 4. DOSA Framework Components—Laboratory Management Data Linked to Rich Soils

### 4.1. Overall Framework

This section briefly describes the entire laboratory data transfer framework using DOSA designed by the core techniques. Figure 6 shows the overall structure of DOSA to enable laboratory-related data to break barriers. The existing laboratory information system is independent. We use the proposed data security protection technology so that data can be stored safely and conditionally. Every laboratory uses the public key algorithm to store encrypted data which can determine ownership. If any supervision departments (such as the Education Bureau) need to collect a certain type of data, they just need to apply for related laboratories. The data owners decide which data can be shared, then find the Education Bureau's public key in DAC that can encrypt related data to the DRC of the Education Bureau so data users can obtain real-time, non-tampered data from various university laboratories. Compared with the current information collected, the DOSA method using the ownership conversion technique can solve existing problems with isolated laboratory data islands.

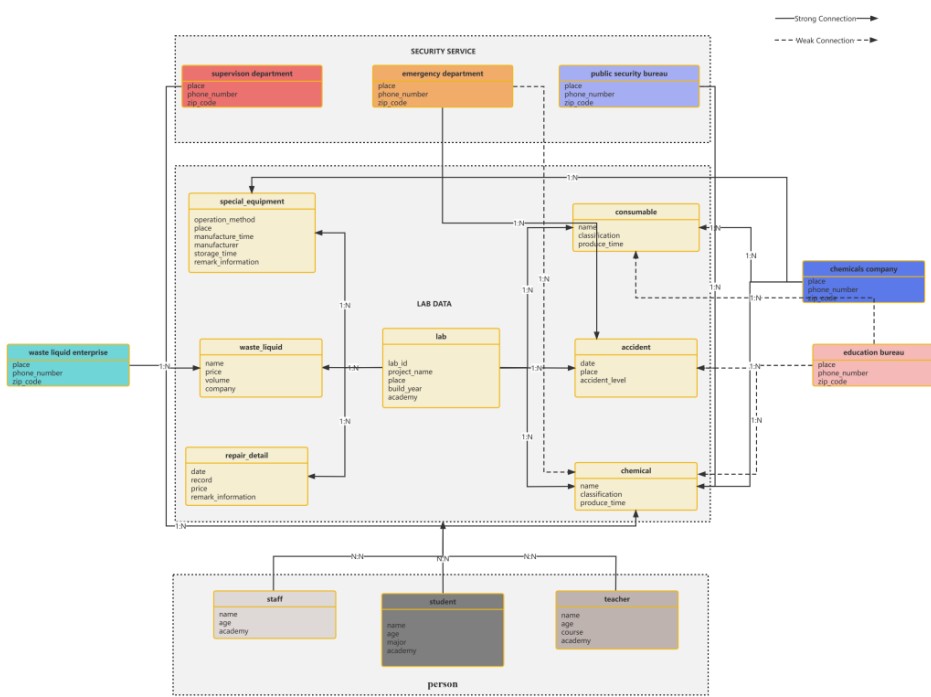

**Figure 6.** The whole structure of laboratory management is based on DOSA.

### 4.2. Laboratory Accident Searching Based on DOSA

We selected the registration and inquiry of laboratory hazardous chemical materials as a case study. The full cycle of hazardous chemical materials includes application, acquirement, procurement, transportation, usage, recycling waste liquid, accidental leakage management, experimental operation accident, etc. The whole process needs to be supervised by multiple relevant units, data need to be circulated, and sharing is needed to break the information cocoon. There are many departments involved, and routine security inspections need to be conducted and results should be submitted according to the protocols, which is not only inefficient but also error-prone and tamper-prone. In case there is an emergency leak or experimental explosion, an integrated and efficient system is necessary to query related data urgently. This workflow of searching for or collecting hazardous chemicals among related branches, such as laboratories, the Education Bureau, the Public Security Bureau Inspection Department, and chemical companies, is shown in Figure 7.

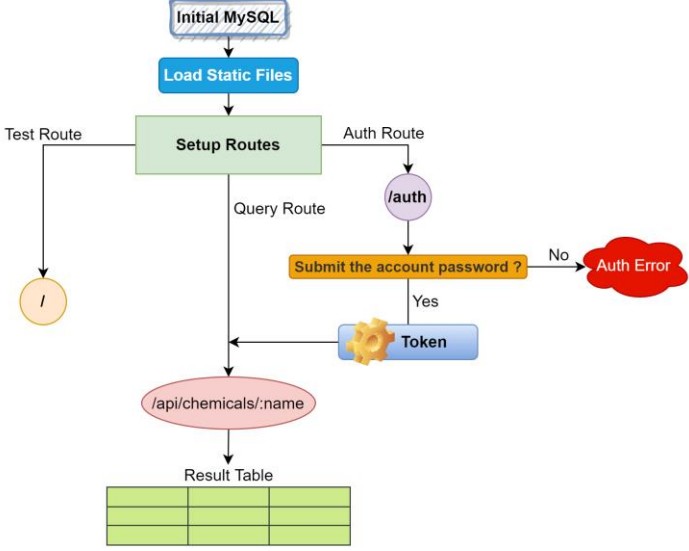

**Figure 7.** The workflow of querying hazardous chemicals.

The hazardous chemicals can be queried through the API specification interface (query by CAS or chemical name) directly among different branches. Hazardous chemical users, administrators, public security departments, emergency departments, and hazardous chemical suppliers can query in real time.

If an accident happens in the laboratory, we could obtain cross-domain and cross-branch information by each unit's authorization in real time. The proposed structure, which can search the corresponding hazardous chemical information, equipment details, relevant personnel profile, and experimental project proposal, etc., is shown in Figure 8.

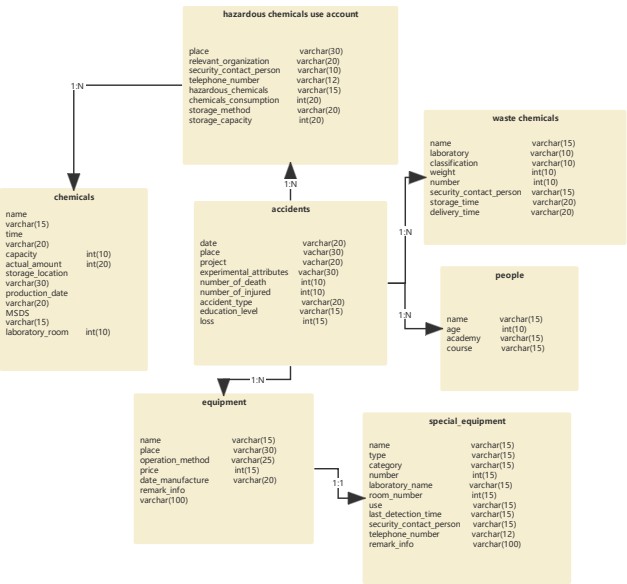

**Figure 8.** Structure of laboratory accident querying in DRC.

## 5. Experiments

In this section, we tested the efficiency of the lab management data based on DOSA by putting the proposed framework into practice. We investigated the time consumption of key generation and public key transmission to DRC by different algorithms using Python 3.10.9. Then, we tried to use four key algorithms, SM2, SM4, RSA, and AES, to compare which algorithm was faster in the proposed architecture regarding authorized encryption and decryption. Combining the security level of the lab data mentioned in the above chapters, we try to find the most suitable solution.

### 5.1. Experimental Environment

The details of experiment environment. The algorithm suits the corresponding experimental data class, which can consider both safety and efficiency (Table 4).

**Table 4.** The details of experiment environment.

| |
| --- |
| Environment 1 HOST 1 |
| Processor: Intel(R) Core (TM) i5-8300H CPU @ 2.30 GHz |
| Memory: 8.0 GB |
| Main hard disk: NVMe WDC PC SN520 SDA |
| Operating system: Windows 11 ×64 Professional Edition Insider Preview |
| Programming software: Microsoft Visual Studio Code |
| Test work data: lab data xlsx. |
| Environment 2 HOST 2 |
| Processor: AMD Ryzen 5 3550H with Radeon Vega Mobile Gfx 2.10 GHz |
| Memory: 8.0 GB |
| Main hard disk: SAMSUNG MZVLB512HAJQ-00000 |
| Operating system: Windows 10 Home |
| Programming software: Pycharm Community Edition |
| Test work data: lab data xlsx. |

### 5.2. Simulation Key Generation and Transmission Experiment

The DAC module includes the right confirmation and authorization. The right confirmation determines the lab data ownership. The ownership of the experimental data belongs to the experimental team designer. After storing data in the DRC, the lab data product adds the research group public key to determine the ownership.

During lab data transactions, the data user wants to purchase the experimental data product. After payment, the data user needs to find the data owner's public key in the DAC, then the user uses the owner's public key to open the lab data product. This is an authorization ownership process.

In this experiment, a key pair was generated in Chengdu, Sichuan Province, China. The public key was sent to Beijing, China (DAC). We carried out 12 experiments generating key pairs, and the tests were performed on env-1 and env-2, respectively. Figure 9 shows a speed test of the time required to generate a key pair. The results of this experiment show that using the AES algorithm is the fastest ($1.232 \times 10^{-5}$ s/$1.605 \times 10^{-5}$ s) in 2 environments. Using SM4 ($3.197 \times 10^{-5}$ s/$6.882 \times 10^{-5}$ s) to create a key pair is the second fastest, which is close to AES. Meanwhile, the SM2 algorithm is the most time-consuming way.

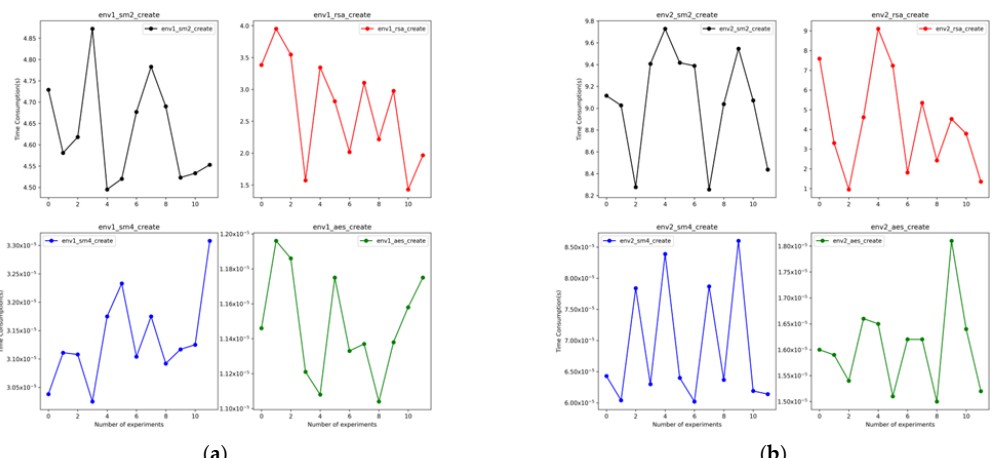

**Figure 9.** Time consumption for generating key pairs using SM2, SM4, AES, and RSA algorithms in env-1 (**a**) and env-2 (**b**).

Further, under two different operating system environments, the experiment simulated the speed of public key transmission from Chengdu to Beijing, and the results are shown in Figure 10. Compared with the key generation experiment, the speed of AES transmission is still the fastest, and the average speed is 1.327 s/1.400 s. In contrast, SM2 is the slowest, with an average speed of 1.485 s/1.571 s.

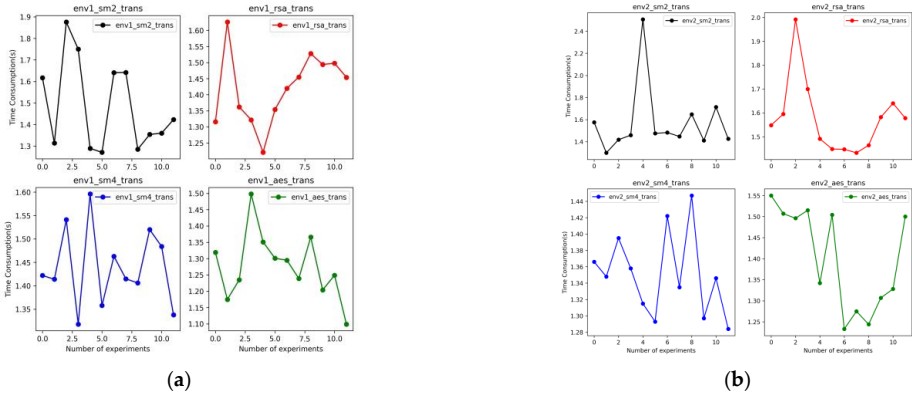

**Figure 10.** Time consumption for simulating transmission public keys to DRC using SM2, SM4, AES, and RSA in env-1 (**a**) and env-2 (**b**).

### 5.3. Confirmation and Authorization Experiments

In the proposed system, we tried to test lab management digital records 50–300 KB, which were encrypted, respectively, using the SM2, SM4, RSA, and AES algorithms, then stored in the DRC of HOST1 and 2. The encryption time of these data texts is shown in Figure 11.

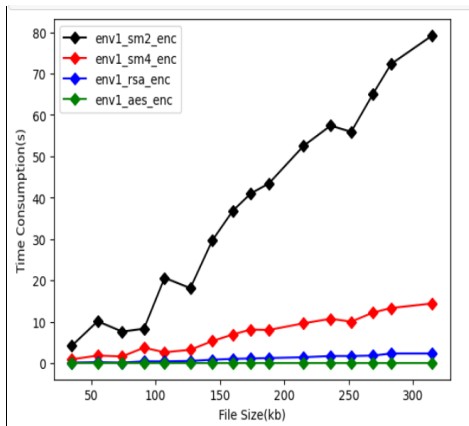
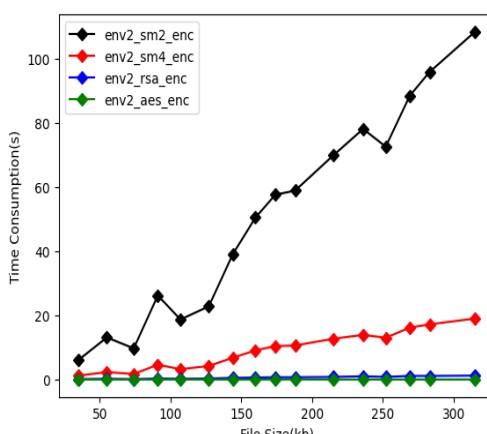

**Figure 11.** Time consumption of authorization encryption of different experimental samples using SM2, SM4, AES, and RSA in env-1 and env-2.

After the lab data user obtained the data owner's public key from DAC, the lab data products were decrypted with four different key management methods, and the time taken is shown in Figure 12. This experiment was performed 16 times, with different sizes of experimental products as variables, under HOST1 and 2. The following is the logic Algorithm 2 for this experiment:

---

**Algorithm 2:** Calculate encryption and decryption time ()

---

Data: example excel file
Result: time consumption for encrypt and decrypt

1.  Data format conversion:
2.     excel file -> json -> string -> bytes
3.  **if** data.type == bytes **then**
4.      time.record
5.      algorithm.encrypt(data)
6.      time.record
7.      algorithm.decrypt(data)
8.      time. record
9.  **end if**
10. calc(time.consumption)

---

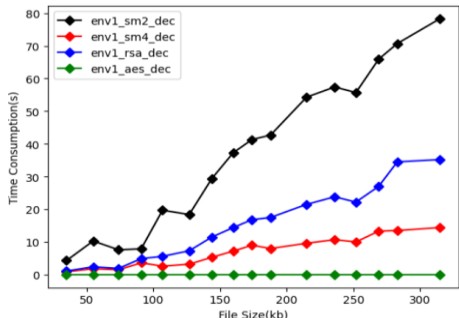
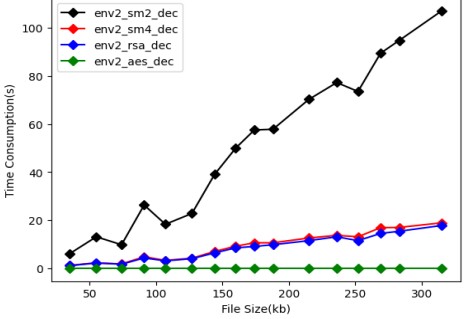

**Figure 12.** Time consumption of authorized decryption of different experimental samples using SM2, SM4, AES, and RSA in env-1 and env-2.

As shown in Figures 11 and 12, AES was the fastest in performance to encrypt and decrypt lab data in the DOSA system. In particular, there are obvious advantages in decrypting data using AES. In Table 2, we have compared the security performance of these four algorithms. AES is a symmetric encryption algorithm, which has the conventional requirement to secretly distribute the key before communication, and the private key must be transmitted to the receiver through the network; thus, the key is not easy to keep secret and manage. However, as we proposed an ownership framework, the private key does not need to be transmitted to the data owner, so this shortcoming can be ignored. In summary, combined with security and speed performance, for lab data of class 1 to 4, it is recommended to use the AES algorithm to encrypt registration to determine ownership using the public key, and to decrypt to authorize with the private key based on DOSA.

However, also need to take high-security measures (class 5 lab data) into account, such as special major laboratory explosion data, or state secret experiment plans. For these security class 5 lab data, we need to pay more attention to safety performance. Combined with Table 2, SM2 shows better safety performance. So, in the current experiment, class 5 was applied to samples to compare the authorization and confirmation time efficiency using SM2.

Host 1's speed of encryption storage of a larger-volume data text (over 230 KB) is double that of Host 2. Additionally, the encryption storage speed of Host 1 for small and medium volume experimental data was three to four times faster than Host 2. However, it seems there is no difference in the data decryption time consumption between the two environments.

As shown in Figure 13, it is recommended to choose a larger experimental sample for class 5 data encryption and decryption, which could save time.

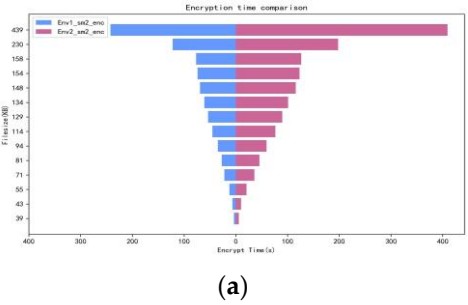
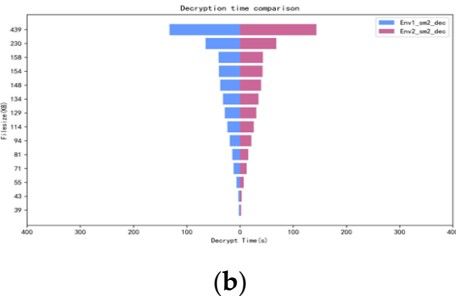

(**a**)  (**b**)

**Figure 13.** Time consumption of encryption (**a**) and decryption (**b**) of different experimental samples using SM2.

*5.4. Evaluate Results*

Combining the above experiments based on the proposed system, it is recommended to use the AES algorithm to encrypt the registration, authorization, and decryption of ordinary security-level data (class 1–4). For a high level of security experimental data (class 5), we recommend using the SM2 algorithm to encrypt larger experimental documents at one time, which can improve the efficiency.

The advantage of DOSA is that it binds data and ownership together, protects data with key technologies, and enables secure sharing. This study designed the whole conditional sharing and transaction process of laboratory-related data based on DOSA. In this process, class 1–5 data are protected by encryption technology to ensure their security. Through the key algorithm and the Gmssl (open-source toolbox), the problem of data leakage can be solved in the data transaction process. Using Python to encrypt the laboratory data with the tools provided by DOSA, it can be considered safe to share relevant laboratory data cross-domain and internally and externally.

## 6. Conclusions

To solve the problem of laboratory data isolated islands, we proposed an ownership-binding person method to confirm lab data rights, and we used a secret key algorithm to implement internal and external laboratory data conditional sharing. A data owner inputs encrypt data into the DRC, while confirming sharing rights. When a data user needs statistics or supervision, he can apply for targeted data by searching the directory. According to the different security levels of lab data, the appropriate key algorithm is selected for ownership authorization, which can realize efficient conditional sharing.

Laboratory data are very necessary for global and cross-border conditional sharing. Often, these data are dispersed across various information systems, posing a challenge in sharing data for a safe and efficient lab data management system. Meanwhile, it is not easy to summarize different sources of real-time data, and it is also hard to search data urgently among the relevant departments of the independent systems. Moreover, the security performance of the different systems is also inconsistent. So, we used data architecture to manage laboratory-related data by allowing relevant units, universities, and laboratories to encrypt data with public keys and upload them to the data register center to form a data directory. Data users can search "Directory" or "data owner" names to apply to the data owner for data viewing or use. Data can be shared conditionally through public key confirmation and private key authorization to break lab data barriers. Then, we used experiments to evaluate which algorithm is better for the efficiency of this system. We suggest AES for ordinary experimental data. For higher levels of experimental data, we can use SM2 to process larger data at one time, which can consider both safety and efficiency. From these experiments, we verified the feasibility and efficient security of DOSA to manage lab-related data.

This lab management data safety ownership architecture can also be applied to other fields, such as smart city construction. The next plan will focus on the prediction of laboratory accidents on this system. Our long-term work goal is to use this data architecture to process the artificial intelligence analysis of experimental management data. Through the entry of a large amount of data, under the premise of the authorization of the data user, we can use the registered massive data to make predictions, such as analyzing the probability of an experimental accident occurring.

**Author Contributions:** Conceptualization, X.Z. and F.M.; methodology, X.Z. and N.C.; software, X.Z.; validation, P.U., X.Z. and F.M.; formal analysis, X.Z. and P.U.; data curation, X.Z. and P.U.; writing—original draft preparation, X.Z.; writing—review and editing, X.Z., P.U. and F.M; visualization, X.Z.; supervision, F.M., N.C. and P.U.; project administration, X.Z.; funding acquisition, X.Z. All authors have read and agreed to the published version of the manuscript.

**Funding:** This research received no external funding.

**Data Availability Statement:** Not applicable.

**Conflicts of Interest:** The authors declare no conflict of interest.

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
