# Peer review of "Registered Data-Centered Lab Management System Based on Data Ownership Safety Architecture"

_electronics, doi:10.3390/electronics12081817_

Round 1
Reviewer 1 Report
The article proposes “… a method to solve the problem of explicit ownership of laboratory management data, conditional sharing, security and efficiency method”.
The research presentation is of low quality: sentences are not connected to each other in the paragraphs, lots of language errors and typos.
Figures 3 -7 are of low quality.
Figure 7 does not have ER participation and cardinality of a relationship.
There are only 7 references.
Line 330. reference Table 2, but Table 2 not presented in the article. These aspects should be addressed at the first place.
Speaking of the research itself it is trivial.
I honestly do not see any reason for it to be published, there is no novelty, no interesting use case, even the experimental results Figure 8-9, 12-13 depends on the different hosts processor speed: HOST1 - Intel(R) Core(TM) i5‐8300H CPU @ 2.30GHz 2.30 GHz, HOST2 - AMD Ryzen 5 3550H with Radeon Vega Mobile Gfx 2.10 GHz.
Author Response
Reviewer: 1 Comments to the Author
- The research presentation is of low quality: sentences are not connected to each other in the paragraphs, lots of language errors and typos.
>> Thanks for your supporting comment, and more contents including experiments were added to this work. Moreover, some English sentences are also be rewritten to revise the grammatical errors, and the detailed responses to your comments one by one as follow.
- Figures 3 -7 are of low quality
>>> Thanks for your kindly reminder. Several better picture was offered to support our paper.
- Figure 7 does not have ER participation and cardinality of a relationship.
>>> Thanks for your valuable comment. We have added ER participation of this Mysql.
- There are only 7 references.
>>> Thanks for your supporting comment, and we have added more references, including DOSA (data ownership safety architecture) application, existing laboratory information management system and key management these references.
- Line 330. reference Table 2, but Table 2 not presented in the article. These aspects should be addressed at the first place.
>>> Thanks for your valuable comment. Our apologies for the careless writing. We have deleted Table 2 because the data is also shown in figures.
- Speaking of the research itself it is trivial.
>>>Thanks for your kindly comment. Laboratory management is really trivial, including reagent, glass consumables, statistical hazardous chemicals etc. We have added the design of whole laboratory management structure to link internal laboratory data sharing with external supervision departments. This would be an excellent direction for future research although addressing this set of questions rigorously is a small and non-trivial undertaking.
- I honestly do not see any reason for it to be published, there is no novelty, no interesting use case, even the experimental results Figure 8-9, 12-13 depends on the different hosts processor speed: HOST1 - Intel(R) Core(TM) i5‐8300H CPU @ 2.30GHz 2.30 GHz, HOST2 - AMD Ryzen 5 3550H with Radeon Vega Mobile Gfx 2.10 GHz.
>>> Thanks for your good comment. We totally agree the experiment result little effect on the designed experimental environment. We have added more experiments and provided some supporting data to evaluate more key algorithm in key generation and transmission.

Reviewer 2 Report
In this paper, the authors presented a method to solve the problem of explicit ownership of laboratory management data, conditional sharing, security, and efficient method.
In order to increase the quality of the paper, the following corrections are recommended:
- At the end of Introduction section, a short description of the paper structure is required.
- Each figure should be cited and explained in the text. Figure 1 is not cited.
- Table 1 should be also cited, and the content should be explained.
- In the second row from table 1 (AES) appear line number (117)
- Section 2 Related work is very briefly presented. Only the results of the 3 authors are presented. The way in which these works are presented is difficult to understand and the authors do not make the connection between the results of these 3 works and the present work.
- Section 3 number appears twice at line 121.
- At line 136, the Methodology section is numbered also with 3.
- Figures 2 (line 156), 3 (line 158), 4 (line 168), and 5 (line 186) are also not cited in the paper.
- Section 4 started directly with 4.1 (line 188). I suppose that this is section 4.
- The abbreviation EXECL (line 201) is not explained. It is not clear what this abbreviation represents.
- In line 259, figure 7 is cited, but my assumption is that figure 9 should be cited here.
- In line 298 figures 10 and 11 should be cited, instead of 9 and 10.
- Table 2 is cited in the paper (line 330), but this table does not exist.
- The References section is too short. The number of works from this section and automatically the number of works cited in this paper is very small. In order to have valuable work, the authors must be aware of the research that has been done in the field.
- A section with a clear description of the comparison between the proposed solution and existing solutions is necessary, as well as a detailed description of the advantages and disadvantages/limitations of the proposed solution.
- The general appearance of the work looks unkempt. I recommend the authors revise its structure. The logical thread of this work is very difficult to follow.

Author Response
Mrs. Ref. No.: electronics-2296314
Title: Registered Data-centered Lab management system based on data ownership safety architecture
Electronics
Dear Editors and Reviewers,
We would like to express our thanks to you and the reviewers for spending time to handle this article. The corrections are carefully performed and major revision is supplied according to the reviewers’ remarks. Thank you.
Best Regards.
Sincerely yours,
Xuying Zheng
International College of Digital Innovation, Chiangmai university, Chiangmai 50200, Thailand.
26th March 2023
Reviewer: 2 Comments to the Author
- At the end of Introduction section, a short description of the paper structure is required.
>>> Thanks for your kindly reminder. We have add more contents to describe paper structure. Section 1 introduce the background. Section 2 explains related theories, and key management, Section 3 presents the core strategies including laboratory data encrypted registration, and authorized transfer method for conditional sharing. Section 4 details the whole lab management framework and chemistry laboratory accident searching based on DOSA using the proposed ownership conversion method. Section 5 verifies the effectiveness of the data flow process using key management. Finally, Section 6 summarizes the paper.
- Each figure should be cited and explained in the text. Figure 1 is not cited.
>>> Thanks for your valuable reminder. We have cited and explained Figure 1 in the text.
- Table 1 should be also cited, and the content should be explained.
>>> Thanks for your valuable reminder. We have cited and explained Table1 in the text.
- In the second row from table 1 (AES) appear line number (117)
>>> Thanks for your valuable comment. Our apologies for careless typesetting, and we have formatted table and line number.
- Section 2 Related work is very briefly presented. Only the results of the 3 authors are presented. The way in which these works are presented is difficult to understand and the authors do not make the connection between the results of these 3 works and the present work.
>>> Thanks for your supporting comment, and we have added more related paper, including existing laboratory information management systems and key management.
- Section 3 number appears twice at line 121.
>>> Our apologies for careless typesetting, and we have deleted Section 3 number.
- At line 136, the Methodology section is numbered also with 3.
>>> Thanks for your valuable comment. Our apologies for careless typesetting. We have formatted Section 3 number.
- Figures 2 (line 156), 3 (line 158), 4 (line 168), and 5 (line 186) are also not cited in the paper.
>>> Thanks for your valuable reminder. We have cited Figure 2,3,4,5 and descried them workflow.
- Section 4 started directly with 4.1 (line 188). I suppose that this is section 4.
>>> Our apologies for careless typesetting, and we have started from Section 4.
- The abbreviation EXECL (line 201) is not explained. It is not clear what this abbreviation represents.
>>> Thanks for your valuable reminder. We have described Execl in text which means chemicals documents .
- In line 259, figure 7 is cited, but my assumption is that figure 9 should be cited here.
>>> Our apologies for careless, and we have changed to figure 9 cited here.
- In line 298 figures 10 and 11 should be cited, instead of 9 and 10.
>>> Our apologies for careless .We have changed figure 9 and 10 which cited here.
- Table 2 is cited in the paper (line 330), but this table does not exist.
>>> Our apologies for careless .We have deleted Table 2 which can shown in Figures.
- The References section is too short. The number of works from this section and automatically the number of works cited in this paper is very small. In order to have valuable work, the authors must be aware of the research that has been done in the field.
>>> Thanks for your supporting comment, and we have added more references, including DOSA (data ownership safety architecture) application, existing laboratory information management system and key management these references.
- A section with a clear description of the comparison between the proposed solution and existing solutions is necessary, as well as a detailed description of the advantages and disadvantages/limitations of the proposed solution.
>>> Thanks for your valuable comment. We have written the table to compare data ownership safety architecture to manage related lab and status laboratory information systems. Then we have described the advantages and limitations of the proposed framework.
- The general appearance of the work looks unkempt. I recommend the authors revise its structure. The logical thread of this work is very difficult to follow.
>>> Thanks for your kindly reminder. We have improved the whole structure.

Round 2
Reviewer 1 Report
The reviewer agrees that the authors have clarified the technical contributions of their work to some extent.
Although several concerns about the paper “Registered Data‐centered Lab management system based on data ownership safety architecture” still need to be solved.
The authors should use typical terms. For example, the authors propose framework (lines 18-19) or system (line 21)? Title of the article “… Lab management system …”. If the authors propose framework they must to view description of the term “framework” and use it carefully.
Unfortunately, the authors did not formulate novelty of the proposed framework or system.
Algorithm 1 and algorithm without number (lines 371-384) provided as painful text. The authors should take into account structure and format of an algorithms (example https://www.mdpi.com/2076-3417/13/7/4168# )
Overall, the authors should polish the readability of the manuscript.
Author Response
Reviewer #1
Thanks a lot for your effort and time to review the present work and provide such valuable and critical comments. Your feedback on the manuscript of the study is highly appreciated by all authors. All of your comments were considered carefully and we provide here an itemized reply to each comment
- The authors should use typical terms. For example, the authors propose framework (lines 1819) or system (line 21)? Title of the article “… Lab management system …”. If the authors propose framework they must to view description of the term “framework” and use it carefully.
Response: Thank you for your comment we have changed “framework” to “system”. in the revised manuscript. Abstract: Line 18, 19, and 21.
- Algorithm 1 and algorithm without number (lines 371-384) provided as painful text. The authors should take into account structure and format of an algorithms 。
We have added line numbers.
- Unfortunately, the authors did not formulate novelty of the proposed framework or system.
Response: Thank you for valuable comment. The novelty statement has been revised according to your instruction in the revised manuscript. Section: 2.2. Laboratory management technology: Line 143-157. Section: 6. Conclusions: Line 406-412.
- The authors should polish the readability of the manuscript.
Response: Thank you for your kind suggestion, the whole manuscript has been carefully revised for the English editing where necessary the modified statement has been blue highlighted in the revised manuscript.

Reviewer 2 Report
The actual structure of the paper is ok.
Author Response
Thank you for your kind suggestion, the whole manuscript has been carefully revised for the English editing where necessary the modified statement has been red highlighted in the revised manuscript.